# The Convergence Rate of Neural Networks for Learned Functions of Different Frequencies

**Ronen Basri**[1]    **David Jacobs**[2]    **Yoni Kasten**[1]    **Shira Kritchman**[1]

[1]Department of Computer Science, Weizmann Institute of Science, Rehovot, Israel
[2]Department of Computer Science, University of Maryland, College Park, MD

## Abstract

We study the relationship between the frequency of a function and the speed at which a neural network learns it. We build on recent results that show that the dynamics of overparameterized neural networks trained with gradient descent can be well approximated by a linear system. When normalized training data is uniformly distributed on a hypersphere, the eigenfunctions of this linear system are spherical harmonic functions. We derive the corresponding eigenvalues for each frequency after introducing a bias term in the model. This bias term had been omitted from the linear network model without significantly affecting previous theoretical results. However, we show theoretically and experimentally that a shallow neural network without bias cannot represent or learn simple, low frequency functions with odd frequencies. Our results lead to specific predictions of the time it will take a network to learn functions of varying frequency. These predictions match the empirical behavior of both shallow and deep networks.

## 1 Introduction

Neural networks have proven effective even though they often contain a large number of trainable parameters that far exceeds the training data size. This defies conventional wisdom that such overparameterization would lead to overfitting and poor generalization. The dynamics of neural networks trained with gradient descent can help explain this phenomenon. If networks explore simpler solutions before complex ones, this would explain why even overparameterized networks settle on simple solutions that do not overfit. It will also imply that early stopping can select simpler solutions that generalize well, [13]. This is demonstrated in Figure 1-left.

We analyze the dynamics of neural networks using a frequency analysis (see also [21, 27, 26, 9], discussed in Section 2). Building on [25, 7, 2] (and under the same assumptions) we show that when a network is trained with a regression loss to learn a function over data drawn from a uniform distribution, it learns the low frequency components of the function significantly more rapidly than the high frequency components (see Figure 2).

Specifically, [7, 2] show that the time needed to learn a function, $f$, is determined by the projection of $f$ onto the eigenvectors of a matrix $H^\infty$, and their corresponding eigenvalues. [25] had previously noted that for uniformly distributed training data, the eigenvectors of this matrix are spherical harmonic functions (analogs to the Fourier basis on hyperspheres). This work makes a number of strong assumptions. They analyze shallow, massively overparameterized networks with no bias. Data is assumed to be normalized.

Building on these results, we compute the eigenvalues of this linear system. Our computation allows us to make specific predictions about how quickly each frequency of the target function will be learned. For example, for the case of 1D functions, we show that a function of frequency $k$ can be

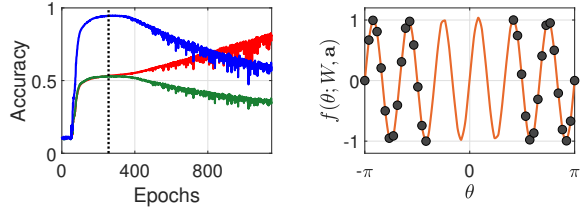

Figure 1: Left: We train a CNN on MNIST data with 50% of the labels randomly changed. As the network trains, accuracy on uncorrupted test data (in blue) first improves dramatically, suggesting that the network first successfully fits the uncorrupted data. Test accuracy then decreases as the network memorizes the incorrectly labeled data. The green curve shows accuracy on test data with mixed correctly/incorrectly labeled data, while the red curve shows training accuracy. (Other papers also mention this phenomenon, e.g., [18]) Right: Given the 1D training data points $(\mathbf{x}_1, ..., \mathbf{x}_{32} \in \mathbb{S}^1)$ marked in black, a two layer network learns the function represented by the orange curve, interpolating the missing data to form an approximate sinusoid of low frequency.

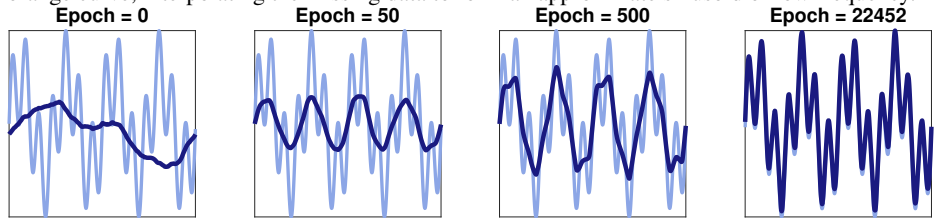

Figure 2: Network prediction (dark blue) for a superposition of two sine waves with frequencies $k = 4, 14$ (light blue). The network fits the lower frequency component of the function after 50 epochs, while fitting the full function only after ∼22K epochs.

learned in time that scales as $k^2$. We show experimentally that this prediction is quite accurate, not only for the simplified networks we study analytically, but also for realistic deep networks.

Bias terms in the network may be neglected without affecting previous theoretical results. However, we show that without bias, two-layer neural networks cannot learn or even represent functions with odd frequencies. This means that in the limit of large data, the bias-free networks studied by [25, 7, 2] cannot learn certain simple, low-frequency functions. We show experimentally that a real shallow network with no bias cannot learn such functions in practice. We therefore modify the model to include bias. We show that with bias added, the eigenvectors remain spherical harmonics, and that odd frequencies can be learned at a rate similar to even frequencies.

Our results show that essentially a network first fits the training data with low frequency functions and then gradually adds higher and higher frequencies to improve the fit. Figure 1-right shows a rather surprising consequence of this. A deep network is trained on the black data points. The orange curve shows the function the network learns. Notice that where there is data missing, the network interpolates with a low frequency function, rather than with a more direct curve. This is because a more straightforward interpolation of the data, while fairly smooth, would contain some high frequency components. The function that is actually learned is almost purely low frequency[1].

This example is rather extreme. In general, our results help to explain why networks generalize well and don't overfit. Because networks learn low frequency functions faster than high frequency ones, if there is a way to fit the data with low-frequency, the network will do this instead of overfitting with a complex, high-frequency function.

## 2 Prior Work

Some prior work has examined the way that the dynamics or architecture of neural networks is related to the frequency of the functions they learn. [21] bound the Fourier transform of the function computed by a deep network and of each gradient descent (GD) update. Their method makes the

strong assumption that the network produces zeros outside a bounded domain. A related analysis for shallow networks is presented in [27, 26]. Neither paper makes an explicit prediction of the speed of convergence. [9] derive bounds that show that for band limited functions two-layer networks converge to a generalizable solution. [20, 24, 8] show that deeper networks can learn high frequency functions that cannot be learned by shallow networks with a comparable number of units. [22] analyzes the ability of networks to learn based on the frequency of functions computed by their components.

Recent papers study the relationship between the dynamics of gradient descent and the ability to generalize. [23] shows that in logistic regression gradient descent leads to max margin solutions for linearly separable data. [5] shows that with the hinge loss a two layer network provably finds a generalizeable solution for linearly separable data. [14, 17] provide related results. [16] studies the effect of gradient descent on the alignment of the weight matrices for linear neural networks. [2] uses the model discussed in this paper to study generalization.

It has been shown that the weights of heavily overparameterized networks change little during training, allowing them to be accurately approximated by linear models that capture the nonlinearities caused by ReLU at initialization [25, 7, 2]. These papers and others analyze neural networks without an explicit bias term [28, 19, 12, 1]. As [1] points out, bias can be ignored without loss of generality for these results, because a constant value can be appended to the training data after it is normalized. [4], building on the work of [3], perform a frequency analysis of the inductive bias of networks, using the Neural Tangent Kernel. They produce results related to ours for bias-free networks. We also analyze the significant effect that bias has on the eigenvalues of these linear systems.

Some recent work (e.g., [6], [12]) raises questions about the relevance of this *lazy training* to practical systems. Interestingly, our experiments indicate that our theoretical predictions, based on lazy training, fit the behavior of real, albeit simple, networks. The relevance of results based on lazy training to large-scale real-world systems remains an interesting topic for future research.

# 3 Background

## 3.1 A Linear Dynamics Model

We begin with a brief review of [7, 2]'s linear dynamics model. We consider a network with two layers, implementing the function

$$f(\mathbf{x}; W, \mathbf{a}) = \frac{1}{\sqrt{m}} \sum_{r=1}^{m} a_r \sigma(\mathbf{w}_r^T \mathbf{x}), \tag{1}$$

where $\mathbf{x} \in \mathbb{R}^{d+1}$ is the input and $\|\mathbf{x}\| = 1$ (denoted $\mathbf{x} \in \mathbb{S}^d$), $W = [\mathbf{w}_1, ..., \mathbf{w}_m] \in \mathbb{R}^{(d+1) \times m}$ and $\mathbf{a} = [a_1, ..., a_m]^T \in \mathbb{R}^m$ respectively are the weights of the first and second layers, and $\sigma$ denotes the ReLU function, $\sigma(x) = \max(x, 0)$. This model does not explicitly include bias. Let the training data consist of $n$ pairs $\{\mathbf{x}_i, y_i\}_{i=1}^n$, $\mathbf{x}_i \in \mathbb{S}^d$ and $y_i \in \mathbb{R}$. Gradient descent (GD) minimizes the $L_2$ loss

$$\Phi(W) = \frac{1}{2} \sum_{i=1}^{n} (y_i - f(\mathbf{x}_i; W, \mathbf{a}))^2, \tag{2}$$

where we initialize the network with $\mathbf{w}_r(0) \sim \mathcal{N}(0, \kappa^2 I)$. We further set $a_r \sim \text{Uniform}\{-1, 1\}$ and maintain it fixed throughout the training.

For the dynamic model we define the $(d+1)m \times n$ matrix

$$Z = \frac{1}{\sqrt{m}} \begin{pmatrix} a_1 \mathbb{I}_{11} \mathbf{x}_1 & a_1 \mathbb{I}_{12} \mathbf{x}_2 & ... & a_1 \mathbb{I}_{1n} \mathbf{x}_n \\ a_2 \mathbb{I}_{21} \mathbf{x}_1 & a_2 \mathbb{I}_{22} \mathbf{x}_2 & ... & a_2 \mathbb{I}_{2n} \mathbf{x}_n \\ ... & & & ... \\ a_m \mathbb{I}_{m1} \mathbf{x}_1 & a_m \mathbb{I}_{m2} \mathbf{x}_2 & ... & a_m \mathbb{I}_{mn} \mathbf{x}_n \end{pmatrix}, \tag{3}$$

where the indicator $\mathbb{I}_{ij} = 1$ if $\mathbf{w}_i^T \mathbf{x}_j \geq 0$ and zero otherwise. Note that this indicator changes from one GD iteration to the next, and so $Z = Z(t)$. The network output over the training data can be expressed as $\mathbf{u}(t) = Z^T \mathbf{w} \in \mathbb{R}^n$, where $\mathbf{w} = (\mathbf{w}_1^T, ..., \mathbf{w}_m^T)^T$. We further define the $n \times n$ Gram matrix $H = H(t) = Z^T Z$ with $H_{ij} = \frac{1}{m} \mathbf{x}_i^T \mathbf{x}_j \sum_{r=1}^{m} \mathbb{I}_{ri} \mathbb{I}_{rj}$.

Next we define the main object of analysis, the $n \times n$ matrix $H^\infty$, defined as the expectation of $H$ over the possible initializations. Its entries are given by

$$H_{ij}^\infty = \mathbb{E}_{\mathbf{w} \sim \mathcal{N}(0, \kappa^2 I)} H_{ij} = \frac{1}{2\pi} \mathbf{x}_i^T \mathbf{x}_j (\pi - \arccos(\mathbf{x}_i^T \mathbf{x}_j)). \qquad (4)$$

Thm. 4.1 in [2] relates the convergence of training a shallow network with GD to the eigenvalues of $H^\infty$. For a network with $m = \Omega\left(\frac{n^7}{\lambda_0^4 \kappa^2 \epsilon^2 \delta}\right)$ units, $\kappa = O\left(\frac{\epsilon \delta}{\sqrt{n}}\right)$ and learning rate $\eta = O\left(\frac{\lambda_0}{n^2}\right)$ ($\lambda_0$ denotes the minimal eigenvalue of $H^\infty$), then with probability $1 - \delta$ over the random initializations

$$\|\mathbf{y} - \mathbf{u}(t)\|_2 = \left(\sum_{i=1}^n (1 - \eta \lambda_i)^{2t} \left(\mathbf{v}_i^T \mathbf{y}\right)^2\right)^{1/2} \pm \epsilon, \qquad (5)$$

where $\mathbf{v}_1, ..., \mathbf{v}_n$ and $\lambda_1, ..., \lambda_n$ respectively are the eigenvectors and eigenvalues of $H^\infty$.

## 3.2 The Eigenvectors of $H^\infty$ for Uniform Data

As is noted in [25], when the training data distributes uniformly on a hypersphere the eigenvectors of $H^\infty$ are the spherical harmonics. In this case $H^\infty$ forms a convolution matrix. A convolution on a hypersphere is defined by

$$K * f(\mathbf{u}) = \int_{\mathbb{S}^d} K(\mathbf{u}^T \mathbf{v}) f(\mathbf{v}) d\mathbf{v}, \qquad (6)$$

where the kernel $K(\mathbf{u}, \mathbf{v}) = K(\mathbf{u}^T \mathbf{v})$ is measureable and absolutely integrable on the hypersphere. It is straightforward to verify that in $\mathbb{S}^1$ this definition is consistent with the standard 1-D convolution with a periodic (and even) kernel, since $K$ depends through the cosine function on the angular difference between $\mathbf{u}$ and $\mathbf{v}$. For $d > 1$ this definition requires the kernel to be rotationally symmetric around the pole. This is essential in order for its rotation on $\mathbb{S}^d$ to make sense. We formalize this observation in a theorem.

**Theorem 1.** *Suppose the training data $\{\mathbf{x}_i\}_{i=1}^n$ is distributed uniformly in $\mathbb{S}^d$, then $H^\infty$ forms a convolution matrix in $\mathbb{S}^d$.*

*Proof.* Let $f : \mathbb{S}^d \to \mathbb{R}$ be a scalar function, and let $\mathbf{f} \in \mathbb{R}^n$ be a vector whose entries are the function values at the training points, i.e., $f_i = f(\mathbf{x}_i)$. Consider the application of $H^\infty$ to $\mathbf{f}$, $g_i = \frac{A(\mathbb{S}^d)}{n} \sum_{j=1}^n H_{ij}^\infty f_j$, where $A(\mathbb{S}^d)$ denotes the total surface area of $\mathbb{S}^d$. As $n \to \infty$ this sum approaches the integral $g(\mathbf{x}_i) = \int_{\mathbb{S}^d} K^\infty(\mathbf{x}_i^T \mathbf{x}_j) f(\mathbf{x}_j) d\mathbf{x}_j$, where $d\mathbf{x}_j$ denotes a surface element of $\mathbb{S}^d$. Let the kernel $K^\infty$ be defined as in (4), i.e., $K^\infty(\mathbf{x}_i, \mathbf{x}_j) = \frac{1}{2\pi} \mathbf{x}_i^T \mathbf{x}_j (\pi - \arccos(\mathbf{x}_i^T \mathbf{x}_j))$. Clearly, $K^\infty$ is rotationally symmetric around $\mathbf{x}_i$, and therefore $g = K^\infty * f$. $H^\infty$ moreover forms a discretization of $K^\infty$, and its rows are phase-shifted copies of each other. $\qquad \square$

Theorem 1 implies that for uniformly distributed data the eigenvectors of $H^\infty$ are the Fourier series in $\mathbb{S}^1$ or, using the Funk-Hecke Theorem (as we will discuss), the spherical harmonics in $\mathbb{S}^d$, $d > 1$. We first extend the dynamic model to allow for bias, and then derive the eigenvalues for both cases.

## 4 Harmonic Analysis of $H^\infty$

These results in the previous section imply that we can determine how quickly a network can learn functions of varying frequency by finding the eigenvalues of the eigenvectors that correspond to these frequencies. In this section we address this problem both theoretically and experimentally[2]. Interestingly, as we establish in Theorem 2 below, the bias-free network defined in (1) is not universal as it cannot represent functions that contain odd frequencies greater than one. As a consequence the odd frequencies lie in the null space of the kernel $K^\infty$ and cannot be learned – a significant deficiency in the model of [7, 2]. We have the following:

**Theorem 2.** *In the harmonic expansion of $f(\mathbf{x})$ in (1), the coefficients corresponding to odd frequencies $k \geq 3$ are zero.*

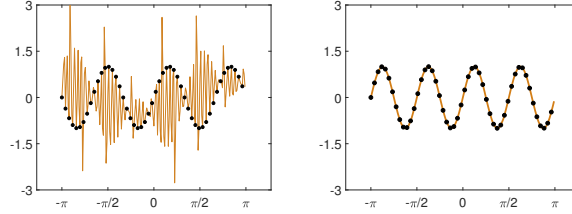

Figure 3: Left: Fitting a bias-free two-layer network (with 2000 hidden units) to training data comprised of 51 points drawn from $f(\theta) = \cos(3\theta)$ (black dots). The orange, solid curve depicts the network output. Consistent with Thm. 2, the network fits the data points perfectly with just even frequencies, yielding poor interpolation between data points. The right panel shows in comparison fitting the network (solid line) to training data points (black dots) drawn from $f(\theta) = \cos(4\theta)$. Fit was achieved by fixing the first layer weights at their random (Gaussian) initialization and optimizing over the second layer weights.

*Proof.* We show this for $d \geq 2$. The theorem also applies to the case that $d = 1$ with a similar proof. Consider the output of one unit, $g(\mathbf{x}) = \sigma(\mathbf{w}^T \mathbf{x})$ and assume first that $w = (0, ..., 0, 1)^T$. In this case $g(\mathbf{x}) = \max\{x_{d+1}, 0\}$ and it is a linear combination of just the zonal harmonics. The zonal harmonic coefficients of $g(x)$ are given by

$$g_k = Vol(\mathbb{S}^{d-1}) \int_{-1}^{1} \max\{t, 0\} P_{k,d}(t)(1 - t^2)^{\frac{d-2}{2}} dt, \tag{7}$$

where $Vol(\mathbb{S}^{d-1})$ denotes the volume of the hypersphere $S^{d-1}$ and $P_{k,d}(t)$ denotes the Gegenbauer polynomial, given by the formula:

$$P_{k,d}(t) = \frac{(-1)^k}{2^k} \frac{\Gamma(\frac{d}{2})}{\Gamma(k + \frac{d}{2})} \frac{1}{(1 - t^2)^{\frac{d-2}{2}}} \frac{d^k}{dt^k}(1 - t^2)^{k + \frac{d-2}{2}}. \tag{8}$$

$\Gamma$ is Euler's gamma function. Eq. (7) can be written as

$$g_k = Vol(\mathbb{S}^{d-1}) \int_{0}^{1} t P_{k,d}(t)(1 - t^2)^{\frac{d-2}{2}} dt. \tag{9}$$

For odd $k$, $P_{k,d}(t)$ is antisymmetric. Therefore, for such $k$

$$g_k = \frac{1}{2} Vol(\mathbb{S}^{d-1}) \int_{1}^{1} t P_{k,d}(t)(1 - t^2)^{\frac{d-2}{2}} dt. \tag{10}$$

This is nothing but the (scaled) inner product of the first order harmonic $t$ with a harmonic of degree $k$, and due to the orthogonality of the harmonic functions this integral vanishes for all odd values of $k$ except $k = 1$. This result remains unchanged if we use a general weight vector for $\mathbf{w}$, as it only rotates $g(\mathbf{x})$, resulting in a phase shift of the first order harmonic. Finally, $f$ is a linear combination of single unit functions, and consequently its harmonic coefficients at odd frequencies $k \geq 3$ are zero. □

In Figure 3 we use a bias-free, two-layer network to fit data drawn from the function $\cos(3\theta)$. Indeed, as the network cannot represent odd frequencies $k \geq 3$ it fits the data points perfectly with combinations of even frequencies, hence yielding poor generalization.

This can be overcome by extending the model to use homogeneous coordinates, which introduce bias. For a point $\mathbf{x} \in \mathbb{S}^d$ we denote $\bar{\mathbf{x}} = \frac{1}{\sqrt{2}}(\mathbf{x}^T, 1)^T \in \mathbb{R}^{d+2}$, and apply (1) to $\bar{\mathbf{x}}$. Clearly, since $\|\mathbf{x}\| = 1$ also $\|\bar{\mathbf{x}}\| = 1$. We note that the proofs of [7, 2] directly apply when both the weights and the biases are initialized using a normal distribution with the same variance. It is also straightforward to modify these theorems to account for bias initialized at zero, as is common in many practical applications. We assume bias is initialized at 0, and construct the corresponding $\bar{H}^\infty$ matrix. This matrix takes the form

$$\bar{H}_{ij}^\infty = \mathbb{E}_{\mathbf{w} \sim \mathcal{N}(0, \kappa^2 I)} \bar{H}_{ij} = \frac{1}{4\pi}(\mathbf{x}_i^T \mathbf{x}_j + 1)(\pi - \arccos(\mathbf{x}_i^T \mathbf{x}_j)). \tag{11}$$

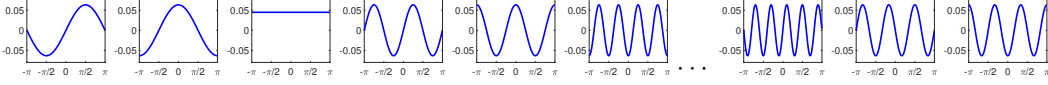

Figure 4: The six leading eigenvectors and three least significant eigenvectors of the bias-free $H^\infty$ in descending order of eigenvalues. Note that the least significant eigenvectors resemble low odd frequencies.

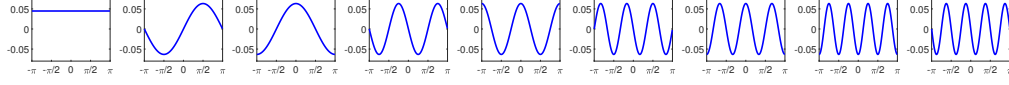

Figure 5: The nine leading eigenvectors ($k = 0, ..., 4$) of $\bar{H}^\infty$ in descending order of eigenvalues. Note that now the leading eigenvectors include both the low even and odd frequencies.

Finally note that the bias adjusted kernel $\bar{K}^\infty(\mathbf{x}_i^T \mathbf{x}_j)$, defined as in (11), also forms a convolution on the original (non-homogeneous) points. Therefore, since we assume that in $\mathbb{S}^d$ the data is distributed uniformly, the eigenfunctions of $\bar{K}^\infty$ are also the spherical harmonics.

We next analyze the eigenfunctions and eigenvalues of $K^\infty$ and $\bar{K}^\infty$. We first consider data distributed uniformly over the circle $\mathbb{S}^1$ and subsequently discuss data in arbitrary dimension.

## 4.1 Eigenvalues in $\mathbb{S}^1$

Since both $K^\infty$ and $\bar{K}^\infty$ form convolution kernels on the circle, their eigenfunctions include the Fourier series. For the bias-free kernel, $K^\infty$, the eigenvalues for frequencies $k \geq 0$ are derived using $a_k^1 = \frac{1}{z_k} \int_{-\pi}^{\pi} K^\infty(\theta) \cos(k\theta) d\theta$ where $z_0 = 2\pi$ and $z_k = \pi$ for $k > 0$. (Note that since $K^\infty$ is an even function its integral with $\sin(\theta)$ vanishes.) This yields

$$
a_k^1 = \begin{cases}
\frac{1}{\pi^2} & k = 0 \\
\frac{1}{4} & k = 1 \\
\frac{2(k^2+1)}{\pi^2(k^2-1)^2} & k \geq 2 \text{ even} \\
0 & k \geq 2 \text{ odd}
\end{cases}
\tag{12}
$$

$H^\infty$ is a discrete matrix that represents convolution with $K^\infty$. It is circulant symmetric (when constructed with points sampled with uniform spacing) and its eigenvectors are real. Each frequency except the DC is represented by two eigenvectors, one for $\sin(k\theta)$ and the other $\cos(k\theta)$.

(12) allows us to make two predictions. First, the eigenvalues for the even frequencies $k$ shrink at the asymptotic rate of $1/k^2$. This suggests, as we show below, that high frequency components are quadratically slower to learn than low frequency components. Secondly, the eigenvalues for the odd frequencies (for $k \geq 3$) vanish. A network without bias cannot learn or even represent these odd frequencies. Du et al.'s convergence results critically depend on the fact that for a finite discretization $H^\infty$ is positive definite. In fact, $H^\infty$ does contain eigenvectors with small eigenvalues that match the odd frequencies on the training data, as shown in Figure 4, which shows the numerically computed eigenvectors of $H^\infty$. The leading eigenvectors include $k = 1$ followed by the low even frequencies, whereas the eigenvectors with smallest eigenvalues include the low odd frequencies. However, a bias-free network can only represent those functions as a combination of even frequencies. These match the odd frequencies on the training data, but have wild behavior off the training data (see Fig. 3). In fact, our experiments show that a network cannot even learn to fit the training data when labeled with odd frequency functions with $k \geq 3$.

With bias, the kernel $\bar{K}^\infty$ passes all frequencies, and the odd frequencies no longer belong to its null space. The Fourier coefficients for this kernel are

$$
c_k^1 = \begin{cases}
\frac{1}{2\pi^2} + \frac{1}{8} & k = 0 \\
\frac{1}{\pi^2} + \frac{1}{8} & k = 1 \\
\frac{k^2+1}{\pi^2(k^2-1)^2} & k \geq 2 \text{ even} \\
\frac{1}{\pi^2 k^2} & k \geq 2 \text{ odd}
\end{cases}
\tag{13}
$$

Figure 5 shows that with bias, the highest eigenvectors include even and odd frequencies.

Thm. 4.1 in [2] tells us how fast a network learning each Fourier component should converge, as a function of the eigenvalues computed in (13). Let $\mathbf{y}_i$ be an eigenvector of $\bar{H}^\infty$ with eigenvalue

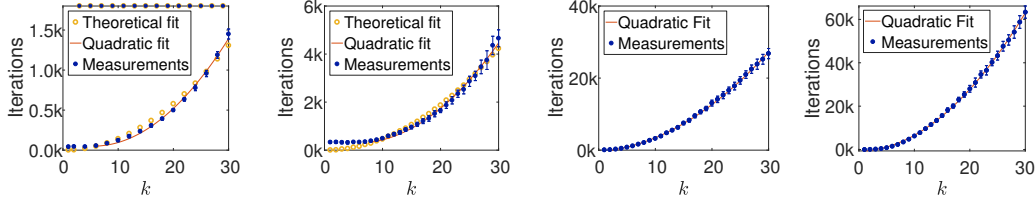

Figure 6: Convergence times as a function of frequency. Left: $\mathbb{S}^1$ no bias ($m = 4000$, $n = 1001$, $\kappa = 1$, $\eta = 0.01$; training odd frequencies was stopped after 1800 iterations had no significant effect on error). Left-center: $\mathbb{S}^1$ with bias ($m = 4000$, $n = 1001$, $\kappa = 2.5$, $\eta = 0.01$). Right-center: deep net (5 hidden layers with bias, $m = 256$, $n = 1001$, $\eta = 0.05$, weight initialized as in [15], bias - uniform). Right: deep residual network (10 hidden layers with same parameters except $\eta = 0.01$). The data lies on a 1D circle embedded in $\mathbb{R}^{30}$ at a random rotation. We estimate the growth in these graphs, from left, as $O(k^{2.15})$, $O(k^{1.93})$, $O(k^{1.94})$, $O(k^{2.11})$. Theoretical predictions (in orange) were scaled by a multiplicative constant to fit the measurements. This constant reflects the length of each gradient step (e.g., due to the learning rate and size of training set). Convergence is declared when a 5% fitting error is obtained.

$\bar{\lambda}_i$ and denote by $t_i$ the number of iterations needed to achieve an accuracy $\bar{\delta}$. Then, according to (5), $(1 - \eta\bar{\lambda}_i)^{t_i} < \bar{\delta} + \epsilon$. Noting that since $\eta$ is small, $\log(1 - \eta\bar{\lambda}_i) \approx -\eta\bar{\lambda}_i$, we obtain that $t_i > \frac{-\log(\bar{\delta}+\epsilon)}{\eta\bar{\lambda}_i}$. Combined with (13) we get that asymptotically in $k$ the convergence time should grow quadratically for *all* frequencies.

We perform experiments to compare theoretical predictions to empirical behavior. We generate uniformly distributed, normalized training data, and assign labels from a single harmonic function. We then train a neural network until the error is reduced to 5% of its original value, and count the number of epochs needed. For odd frequencies and bias-free 2-layer networks we halt training when the network fails to significantly reduce the error in a large number of epochs. We run experiments with shallow networks and with deep fully connected networks and deep networks with skip connections. We primarily use an $L_2$ loss, but in supplementary material we show results with a cross-entropy loss. Quadratic behavior is observed in all these cases, see Figure 6. The actual convergence times may vary with the details of the architecture and initialization. For very low frequencies the run time is affected more strongly by the initialization, yielding slightly slower convergence times than predicted.

Thm. 5.1 in [2] further allows us to bound the generalization error incurred when learning band limited functions. Suppose $\mathbf{y} = \sum_{k=0}^{\bar{k}} \alpha_k e^{2\pi i k x}$. According to this theorem, and noting that the eigenvalues of $(\bar{H}^\infty)^{-1} \approx \pi k^2$, with sufficiently many iterations the population loss $L_\mathcal{D}$ computed over the entire data distribution is bounded by

$$L_\mathcal{D} \lesssim \sqrt{\frac{2\mathbf{y}(\bar{H}^\infty)^{-1}\mathbf{y}}{n}} \approx \sqrt{\frac{2\pi \sum_{k=1}^{\bar{k}} \alpha_k^2 k^2}{n}}. \tag{14}$$

As expected, the lower the frequency is, the lower the generalization bound is. For a pure sine wave the bound increases linearly with frequency $k$.

## 4.2 Eigenvalues in $\mathbb{S}^d, d \geq 2$

To analyze the eigenvectors of $H^\infty$ when the input is higher dimensional, we must make use of generalizations of the Fourier basis and convolution to functions on a high dimensional hypersphere. Spherical harmonics provide an appropriate generalization of the Fourier basis (see [11] as a reference for the following discussion). As with the Fourier basis, we can express functions on the hypersphere as linear combinations of spherical harmonics. Since the kernel is rotationally symmetric, and therefore a function of one variable, it can be written as a linear combination of the *zonal* harmonics. For every frequency, there is a single zonal harmonic which is also a function of one variable. The zonal harmonic is given by the Gegenbauer polynomial, $P_{k,d}$ where $k$ denotes the frequency, and $d$ denotes the dimension of the hypersphere.

We have already defined convolution in (6) in a way that is general for convolution on the hypersphere. The Funk-Hecke theorem provides a generalization of the convolution theorem for spherical harmonics, allowing us to perform a frequency analysis of the convolution kernel. It states:

**Theorem 3.** *(Funk-Hecke) Given any measurable function $K$ on $[-1, 1]$, such that the integral: $\int_{-1}^{1} \|K(t)\|(1-t^2)^{\frac{d-2}{2}} dt < \infty$, for every spherical harmonic $H(\sigma)$ of frequency $k$, we have:*

$$\int_{\mathbb{S}^d} K(\sigma \cdot \xi) H(\xi) d\xi = \left( \text{Vol}(\mathbb{S}^{d-1}) \int_{-1}^{1} K(t) P_{k,d}(t)(1-t^2)^{\frac{d-2}{2}} dt \right) H(\sigma).$$

Here $\text{Vol}(\mathbb{S}^{d-1})$ denotes the volume of $\mathbb{S}^{d-1}$ and $P_{k,d}(t)$ denotes the Gegenbauer polynomial defined in (8). This tells us that the spherical harmonics are the eigenfunctions of convolution. The eigenvalues can be found by taking an inner product between $K$ and the zonal harmonic of frequency $k$. Consequently, we see that for uniformly distributed input, in the limit for $n \to \infty$, the eigenvectors of $H^\infty$ are the spherical harmonics in $\mathbb{S}^d$.

Similar to the case of $\mathbb{S}^1$, in the bias free case the odd harmonics with $k \geq 3$ lie in the null space of $K^\infty$. This is proved in the following theorem.

**Theorem 4.** *The eigenvalues of convolution with $K^\infty$ vanish when they correspond to odd harmonics with $k \geq 3$.*

*Proof.* Consider the vector function $\mathbf{z}(\mathbf{w}, \mathbf{x}) = \mathbb{I}(\mathbf{w}^T \mathbf{x} > 0)\mathbf{x}$ and note that $K^\infty(\mathbf{x}_i, \mathbf{x}_j) = \int_{\mathbb{S}^d} \mathbf{z}^T(\mathbf{w}, \mathbf{x}_i)\mathbf{z}(\mathbf{w}, \mathbf{x}_j) d\mathbf{w}$. Let $y(\mathbf{x})$ be an odd order harmonic of frequency $k > 1$. The application of $\mathbf{z}$ to $y$ takes the form

$$\int_{\mathbb{S}^d} \mathbf{z}(\mathbf{w}, \mathbf{x}) y(\mathbf{x}) d\mathbf{x} = \int_{\mathbb{S}^d} \mathbb{I}(\mathbf{w}^T \mathbf{x} > 0) \mathbf{g}(\mathbf{x}) d\mathbf{x}, \tag{15}$$

where $\mathbf{g}(\mathbf{x}) = y(\mathbf{x})\mathbf{x}$. $\mathbf{g}(\mathbf{x})$ is a $(d+1)$-vector whose $l^{\text{th}}$ coordinate is $g^l(\mathbf{x}) = x^l y(x)$. We first note that $g^l(\mathbf{x})$ has no DC component. This is because $g^l$ is the product of two harmonics, the scaled first order harmonic, $x^l$, and the odd harmonic $y(\mathbf{x})$ (with $k > 1$), so their inner product vanishes.

Next we will show that the kernel $\mathbb{I}(\mathbf{w}^T \mathbf{x} > 0)$ annihilates the even harmonics, for $k > 1$. Note that the odd/even harmonics can be written as a sum of monomials of odd/even degrees. Since $g$ is the sum of even harmonics (the product of $x^l$ and an odd harmonic) this will imply that (15) vanishes. Using the Funk-Hecke theorem, the even coefficients of the kernel (with $k > 1$) are

$$
\begin{aligned}
r_k^d &= \text{Vol}(\mathbb{S}^{d-1}) \int_{-1}^{1} \mathbb{I}(t > 0) P_{k,d}(t)(1-t^2)^{\frac{d-2}{2}} dt \tag{16} \\
&= \text{Vol}(\mathbb{S}^{d-1}) \int_{0}^{1} P_{k,d}(t)(1-t^2)^{\frac{d-2}{2}} dt = \frac{\text{Vol}(\mathbb{S}^{d-1})}{2} \int_{-1}^{1} P_{k,d}(t)(1-t^2)^{\frac{d-2}{2}} dt = 0.
\end{aligned}
$$

When we align the kernel with the zonal harmonic, $\mathbf{w}^T \mathbf{x} = t$, justifying the second equality. The third equality is due to the symmetry of the even harmonics, and the last equality is because the harmonics of $k > 0$ are zero mean. $\square$

Next we compute the eigenvalues of both $K^\infty$ and $\bar{K}^\infty$ (for simplicity we show only the case of even $d$, see supplementary material for the calculations). We find for networks without bias:

$$a_k^d = \begin{cases} C_1(d, 0)\frac{1}{d2^{d+1}}\binom{d}{2} & k = 0 \\ C_1(d, 1)\sum_{q=1}^{d} C_2(q, d, 1)\frac{1}{2(2q+1)} & k = 1 \\ C_1(d, k)\sum_{q=\lceil \frac{k}{2}\rceil}^{k+\frac{d-2}{2}} C_2(q, d, k)\frac{1}{2(2q-k+2)}\left(1 - \frac{1}{2^{2q-k+2}}\binom{2q-k+2}{\frac{2q-k+2}{2}}\right) & k \geq 2 \text{ even} \\ 0 & k \geq 2 \text{ odd,} \end{cases} \tag{17}$$

with

$$C_1(d, k) = \frac{\pi^{\frac{d}{2}}}{(\frac{d}{2})}\frac{(-1)^k}{2^k}\frac{1}{\Gamma(k + \frac{d}{2})}, \qquad C_2(q, d, k) = (-1)^q \binom{k + \frac{d-2}{2}}{q}\frac{(2q)!}{(2q-k)!}.$$

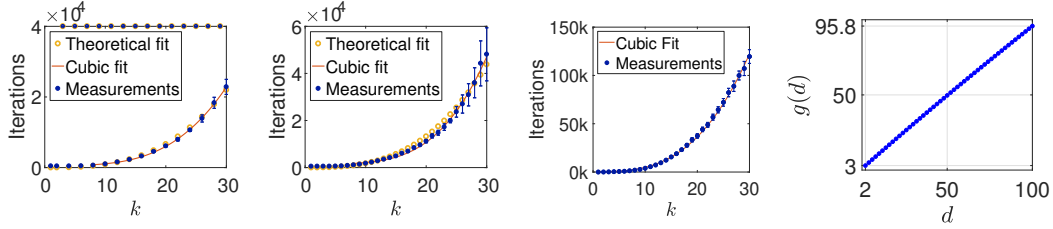

Figure 7: Convergence times as a function of frequency for data in $\mathbb{S}^2$. Left: no bias ($m = 16000$, $n = 1001$, $\kappa = 1$, and $\eta = 0.01$; training odd frequencies was stopped after 40K iterations with no significant reduction of error). Left-center: with bias (same parameters). Right-center: deep residual network (10 hidden layers with $m = 256$, $n = 5000$, $\eta = 0.001$, weight initialization as in [15], bias - uniform). The data lies on a 2D sphere embedded in $\mathbb{R}^{30}$ at a random rotation. Growth estimates from left, $O(k^{2.74}), O(k^{2.87}), O(k^{3.13})$. Right: Convergence exponent as a function of dimension. $g(d) = \lim_{k \to \infty} -\frac{\log c_k^d}{\log k}$ estimated by calculating the coefficients up to $k = 1000$, indicating that the coefficients decay roughly as $1/k^d$.

Adding bias to the network, the eigenvalues for $\bar{K}^\infty$ are:

$$
c_k^d = \begin{cases}
\frac{1}{2}C_1(d,0)\left(\frac{1}{d2^{d+1}}\binom{d}{\frac{d}{2}} + \frac{2^{d-1}}{d\binom{d-1}{\frac{d}{2}}} - \frac{1}{2}\sum_{q=0}^{\frac{d-2}{2}}(-1)^q\binom{\frac{d-2}{2}}{q}\frac{1}{2q+1}\right) & k = 0 \\
\frac{1}{2}C_1(d,1)\sum_{q=\lceil\frac{k}{2}\rceil}^{k+\frac{d-2}{2}}C_2(q,d,1)\left(\frac{1}{2(2q+1)} + \frac{1}{4q}\left(1 - \frac{1}{2^{2q}}\binom{2q}{q}\right)\right) & k = 1 \\
\frac{1}{2}C_1(d,k)\sum_{q=\lceil\frac{k}{2}\rceil}^{k+\frac{d-2}{2}}C_2(q,d,k)\left(\frac{-1}{2(2q-k+1)} + \frac{1}{2(2q-k+2)}\left(1 - \frac{1}{2^{2q-k+2}}\binom{2q-k+2}{\frac{2q-k+2}{2}}\right)\right) & k \geq 2 \text{ even} \\
\frac{1}{2}C_1(d,k)\sum_{q=\lceil\frac{k}{2}\rceil}^{k+\frac{d-2}{2}}C_2(q,d,k)\left(\frac{1}{2(2q-k+1)}\left(1 - \frac{1}{2^{2q-k+1}}\binom{2q-k+1}{\frac{2q-k+1}{2}}\right)\right) & k \geq 2 \text{ odd.}
\end{cases}
\tag{18}
$$

We trained two layer networks with and without bias, as well as a deeper network, on data representing pure spherical harmonics in $\mathbb{S}^2$. Convergence times are plotted in Figure 7. These times increase roughly as $k^3$, matching our predictions in (17) and (18). We further estimated numerically the anticipated convergence times for data of higher dimension. As the figure shows (right panel), convergence times are expected to grow roughly as $k^d$. We note that this is similar to the bound derived in [21] under quite different assumptions.

## 5 Discussion

We have developed a quantitative understanding of the speed at which neural networks learn functions of different frequencies. This shows that they learn high frequency functions much more slowly than low frequency functions. Our analysis addresses networks that are heavily overparameterized, but our experiments suggest that these results apply to real neural networks.

This analysis allows us to understand gradient descent as a frequency based regularization. Essentially, networks first fit low frequency components of a target function, then they fit high frequency components. This suggests that early stopping regularizes by selecting smoother functions. It also suggests that when a network can represent many functions that would fit the training data, gradient descent causes the network to fit the smoothest function, as measured by the power spectrum of the function. In signal processing, it is commonly the case that the noise contains much larger high frequency components than the signal. Hence smoothing reduces the noise while preserving most of the signal. Gradient descent may perform a similar type of smoothing in neural networks.

**Acknowledgments**. The authors thank Adam Klivans, Boaz Nadler, and Uri Shaham for helpful discussions. This material is based upon work supported by the National Science Foundation under Grant No. DMS1439786 while the authors were in residence at the Institute for Computational and Experimental Research in Mathematics in Providence, RI, during the Computer Vision program. This research is supported by the National Science Foundation under grant no. IIS-1526234.

## Footnotes

[1] [10] show a related figure. In the context of meta-learning they show that a network trained to regress to sine waves can learn a new sine wave from little training data. Our figure shows a different phenomenon, that, when possible, a generic network will fit data with low-frequency sine waves.

[2]Code for experiments shown in this paper can be found at https://github.com/ykasten/Convergence-Rate-NN-Different-Frequencies.

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
