[Supplementary Material · Supplementary Material (Final).pdf]

# The Convergence Rate of Neural Networks for Learned Functions of Different Frequencies Supplementary Material

**Ronen Basri**[1]  **David Jacobs**[2]  **Yoni Kasten**[1]  **Shira Kritchman**[1]

[1]Department of Computer Science, Weizmann Institute of Science, Rehovot, Israel
[2]Department of Computer Science, University of Maryland, College Park, MD

## A  Cross entropy loss

While this is outside the scope of the theoretical results in the paper, we tested the convergence rate of a network with a single hidden layer with the cross entropy loss. We used a binary classification task. To construct our target classes, for every integer $k > 0$ we produced data on the 1D circle according to the function $\cos(k\theta)$, and then thresholded it, assigning class 1 if $\cos(k\theta) > 2/3$, -1 if $\cos(k\theta) < 2/3$ and omitted points for which $|\cos(k\theta)| \leq 2/3$. As with the MSE loss, here too we see a near quadratic convergence rate, see Figure 1.

Figure 1: Number of iterations to convergence as a function of target frequency with the cross entropy loss. A deep residual network is used with 10 hidden layers including bias, $m = 256$, $\eta = 0.05$, $n = 1001$. Weight initialization as in [13], bias - uniform. Leading exponent is estimated as $O(K^{2.34})$.

## B  Eigenvalues of $H^\infty$ with $d > 1$

Using the Funk-Hecke theorem, we can find the eigenvalues of $H^\infty$ in the continuous limit by integrating the product of the convolution kernel with spherical harmonics. We first collect together a number of formulas and integrals that will be useful. We then show how to use the Funk-Hecke theorem to formulate the relevant integrals, and finally compute the results.

### B.1  Useful integrals and equations

$\int_0^\pi \cos^n \theta d\theta$ is $\pi$ for $n = 0$ and 0 for $n = 1$. For $n > 1$ we use integration by parts

$$\int_0^\pi \cos^n \theta d\theta = \left. \frac{\cos^{n-1}\theta \sin\theta}{n} \right|_0^\pi + \frac{n-1}{n} \int_0^\pi \cos^{n-2} \theta d\theta. \tag{1}$$

The first term vanishes and we obtain

$$\int_0^\pi \cos^n \theta d\theta = \begin{cases} \pi \frac{n-1}{n} \frac{n-3}{n-2} \cdots \frac{1}{2} = \frac{\pi}{2^n} \binom{n}{\frac{n}{2}} & n \text{ is even.} \\ 0 & n \text{ is odd} \end{cases} \tag{2}$$

$\int_0^\pi \sin^n \theta d\theta$ is $\pi$ for $n = 0$ and 2 for $n = 1$. For $n > 1$ we integrate by parts

$$\int_0^\pi \sin^n \theta d\theta = \left. \frac{-\sin^{n-1}\theta \cos\theta}{n} \right|_0^\pi + \frac{n-1}{n} \int_0^\pi \sin^{n-2}\theta d\theta. \tag{3}$$

The first term vanishes, and we obtain

$$\int_0^\pi \sin^n \theta d\theta = \begin{cases} \pi \frac{n-1}{n} \frac{n-3}{n-2} \cdots \frac{1}{2} = \frac{\pi}{2^n}\binom{n}{\frac{n}{2}} & n \text{ is even.} \\ 2\frac{n-1}{n} \frac{n-3}{n-2} \cdots \frac{2}{3} = \frac{2^{n+1}}{(n+1)\left(\frac{n+1}{2}\right)} & n \text{ is odd} \end{cases} \tag{4}$$

Next we wish to compute $\int_0^\pi \theta \cos^n \theta \sin \theta d\theta$ for $n \geq 1$. Integrating by parts

$$\int_0^\pi \theta \cos^n \theta \sin \theta d\theta = \left. -\frac{\theta \cos^{n+1}\theta}{n+1} \right|_0^\pi + \int_0^\pi \frac{\cos^{n+1}\theta}{n+1} d\theta. \tag{5}$$

Using (2) this we obtain

$$\begin{aligned}
\int_0^\pi \theta \cos^n \theta \sin \theta d\theta &= \frac{(-1)^n \pi}{n+1} + \begin{cases} 0 & n \text{ is even} \\ \frac{\pi}{n+1} \frac{n}{n+1} \frac{n-2}{n-1} \cdots \frac{1}{2} & n \text{ is odd} \end{cases} \\
&= \begin{cases} \frac{\pi}{n+1} & n \text{ is even.} \\ \frac{\pi}{n+1}\left(-1 + \frac{1}{2^{n+1}}\binom{n+1}{\frac{n+1}{2}}\right) & n \text{ is odd} \end{cases}
\end{aligned} \tag{6}$$

Next

$$\int_0^\pi \theta \cos\theta \sin^n \theta d\theta = \left. \frac{\theta \sin^{n+1}\theta}{n+1} \right|_0^\pi - \int_0^\pi \frac{\sin^{n+1}\theta}{n+1} d\theta \tag{7}$$

The first term vanishes and we obtain from (4)

$$\int_0^\pi \theta \cos\theta \sin^n \theta d\theta = \begin{cases} -\frac{2^{n+2}}{(n+1)(n+2)\binom{n+1}{\frac{n+2}{2}}} & n \text{ is even} \\ -\frac{\pi}{(n+1)2^{n+1}}\binom{n+1}{\frac{n+1}{2}} & n \text{ is odd.} \end{cases} \tag{8}$$

Other useful equations

$$(1 - t^2)^p = \sum_{q=0}^p (-1)^q \binom{p}{q} t^{2q} \tag{9}$$

and its $k$'th derivative,

$$\frac{d^k}{dt^k}(1 - t^2)^p = \sum_{q=\lceil \frac{k}{2} \rceil}^p C_2(q, d, k) t^{2q-k} \tag{10}$$

where we denote

$$C_2(q, d, k) = (-1)^q \binom{p}{q} \frac{(2q)!}{(2q-k)!} \tag{11}$$

$$\int_{-1}^1 t^n dt = \left. \frac{t^{n+1}}{n+1} \right|_{-1}^1 = \frac{1 - (-1)^{n+1}}{n+1} = \begin{cases} \frac{2}{n+1} & n \text{ is even} \\ 0 & n \text{ is odd} \end{cases} \tag{12}$$

$$\int_{-1}^1 t(1 - t^2)^n dt = 0, \tag{13}$$

since this is a product of an odd and even functions.

Finally, using (6) and (9),

$$\begin{aligned}
\int_{-1}^1 \arccos(t)(1 - t^2)^n dt &= \sum_{q=0}^n (-1)^q \binom{n}{q} \int_0^\pi \theta \cos^{2q}\theta \sin\theta d\theta \\
&= \sum_{q=0}^n (-1)^q \binom{n}{q} \frac{\pi}{2q+1}
\end{aligned} \tag{14}$$

## B.2 The Kernel

We have

$$H_{i,j}^\infty = \frac{t(\pi - \arccos(t))}{2\pi} = \frac{\cos\theta(\pi - |\theta|)}{2\pi} \tag{15}$$

for $\theta$ the angle between $x_i$ and $x_j$ and we use the notation $t = \cos\theta$. For the case of $x_i$ uniformly sampled on the hypersphere, this amounts to convolution by the kernel:

$$K^\infty = \frac{\pi\cos\theta - \theta\cos\theta}{2\pi} \tag{16}$$

The absolute value disappears because on the hypersphere, $\theta$ varies between $0$ and $\pi$.

For the bias, the kernel changes to

$$\bar{K}^\infty = \frac{(t+1)(\pi - \arccos(t))}{4\pi} = \frac{(\cos\theta + 1)(\pi - \theta)}{4\pi} \tag{17}$$

We can divide the integrals we need to compute into four parts. We denote:

$$K_1 = \frac{t}{2} = \frac{\cos\theta}{2} \tag{18}$$

$$K_2 = -\frac{t\arccos(t)}{2\pi} = -\frac{\theta\cos\theta}{2\pi} \tag{19}$$

$$K_3 = \frac{1}{2} \tag{20}$$

$$K_4 = -\frac{\arccos(t)}{2\pi} = -\frac{\theta}{2\pi} \tag{21}$$

This gives us $K^\infty = K_1 + K_2$. We denote $K^b = K_3 + K_4$. This is the new component introduced by bias. Then we have $\bar{K}^\infty = \frac{1}{2}(K^\infty + K^b) = \frac{1}{2}(K_1 + K_2 + K_3 + K_4)$. We will use $a_k^d$ to denote the coefficient for frequency $k$ of the harmonic transform of $K^\infty$, in dimension $d$. We use $b_k^d$ to denote the coefficient of the transform for just the bias term, $K^b$. And finally, $c_k^d$ denotes the coefficient for the complete kernel with bias, $\bar{K}^\infty$, so that $c_k^d = a_k^d + b_k^d$.

## B.3 Application of the Funk Hecke theorem

The eigenvalues of $H^\infty$ can be found by projecting the kernel onto the spherical harmonics, that is, by taking their transform. It is only necessary to do this for the zonal harmonics. This is because the kernel is written so that it only has components in the zonal harmonic. Suppose the dimension of $x_i$ is $d+1$, so it lies on $\mathbb{S}^d$, and we want to compute the transform for the $k$'th order harmonic. We have:

$$a_k^d = Vol(\mathbb{S}^{d-1})\int_{-1}^{1} K^\infty(t)P_{k,d}(t)(1-t^2)^{\frac{d-2}{2}}\,dt, \tag{22}$$

where $Vol(\mathbb{S}^{d-1})$ denotes the volume of $S^{d-1}$, given by

$$Vol(\mathbb{S}^{d-1}) = \frac{\pi^{\frac{d}{2}}}{\Gamma(\frac{d}{2}+1)} \tag{23}$$

and $P_{k,d}(t)$ denotes the Gegenbauer polynomial, given by the formula:

$$P_{k,d}(t) = \frac{(-1)^k}{2^k}\frac{\Gamma(\frac{d}{2})}{\Gamma(k+\frac{d}{2})}\frac{1}{(1-t^2)^{\frac{d-2}{2}}}\frac{d^k}{dt^k}(1-t^2)^{k+\frac{d-2}{2}} \tag{24}$$

$\Gamma$ is Euler's gamma function whose formulas for integer values of $n$ are:

$$\Gamma(n) = (n-1)! \tag{25}$$

$$\Gamma(n+\frac{1}{2}) = (n-\frac{1}{2})(n-\frac{3}{2})...\frac{1}{2}\pi^{\frac{1}{2}} \tag{26}$$

Substituting for these terms we obtain

$$
\begin{aligned}
a_k^d &= \frac{\pi^{\frac{d}{2}}}{\Gamma(\frac{d}{2}+1)} \int_{-1}^{1} K^\infty(t) \frac{(-1)^k}{2^k} \frac{\Gamma(\frac{d}{2})}{\Gamma(k+\frac{d}{2})} \frac{1}{(1-t^2)^{\frac{d-2}{2}}} \\
&\quad \left(\frac{d^k}{dt^k}(1-t^2)^{k+\frac{d-2}{2}}\right)(1-t^2)^{\frac{d-2}{2}} dt & (27) \\
&= \frac{\pi^{\frac{d}{2}}}{\Gamma(\frac{d}{2}+1)} \int_{-1}^{1} K^\infty(t) \frac{(-1)^k}{2^k} \frac{\Gamma(\frac{d}{2})}{\Gamma(k+\frac{d}{2})} \frac{d^k}{dt^k}(1-t^2)^{k+\frac{d-2}{2}} dt & (28) \\
&= \frac{\pi^{\frac{d}{2}}}{\Gamma(\frac{d}{2}+1)} \frac{(-1)^k}{2^k} \frac{\Gamma(\frac{d}{2})}{\Gamma(k+\frac{d}{2})} \int_{-1}^{1} K^\infty(t) \frac{d^k}{dt^k}(1-t^2)^{k+\frac{d-2}{2}} dt & (29) \\
&= C_1(d,k) \int_{-1}^{1} K^\infty(t) \frac{d^k}{dt^k}(1-t^2)^{k+\frac{d-2}{2}} dt & (30) \\
& & (31)
\end{aligned}
$$

with

$$
C_1(d,k) = \frac{\pi^{\frac{d}{2}}}{(\frac{d}{2})} \frac{(-1)^k}{2^k} \frac{1}{\Gamma(k+\frac{d}{2})}
$$

To simplify the expressions we obtain, we will assume $d$ is even in what follows. For the cases with and without bias we first compute the DC component of the parts of the kernels, and then compute the coefficients for $k > 0$.

## B.4 Calculating the coefficients: no bias

__k = 0__:

$$
a_0^d = C_1(d,k) \int_{-1}^{1} K^\infty(t)(1-t^2)^{\frac{d-2}{2}} dt. \tag{32}
$$

First we consider $K_1$ (18). Using (13) we have

$$
\frac{1}{2} \int_{-1}^{1} t(1-t^2)^{\frac{d-2}{2}} dt = 0. \tag{33}
$$

Next, we consider $K_2$ (19). Using (8) we have

$$
-\frac{1}{2\pi} \int_0^\pi \theta \cos\theta \sin^{d-1}\theta d\theta = \frac{1}{d2^{d+1}} \binom{d}{\frac{d}{2}} \tag{34}
$$

Therefore,

$$
a_0^d = C_1(d,k) \frac{1}{d2^{d+1}} \binom{d}{\frac{d}{2}} \tag{35}
$$

__k > 0__:

$$
a_k^d = C_1(d,k) \int_{-1}^{1} K^\infty(t) \frac{d^k}{dt^k}(1-t^2)^p dt \tag{36}
$$

where we denote $p = k + \frac{d-2}{2}$, noting that $p \geq k$. Using (10)

$$
a_k^d = C_1(d,k) \sum_{q=\lceil \frac{k}{2} \rceil}^{p} C_2(q,d,k) \int_{-1}^{1} K^\infty(t) t^{2q-k} dt \tag{37}
$$

Considering $K_1$, and using (12)

$$
\frac{1}{2} \int_{-1}^{1} t^{2q-k+1} dt = \begin{cases} 0 & k \text{ is even} \\ \frac{1}{2q-k+2} & k \text{ is odd} \end{cases} \tag{38}
$$

Considering $K_2$, and using (6)

$$\frac{1}{2\pi} \int_0^\pi \theta \cos^{2q-k+1}\theta \sin\theta d\theta = \begin{cases} \frac{1}{2(2q-k+2)}\left(-1 + \frac{1}{2^{2q-k+2}}\binom{2q-k+2}{\frac{2q-k+2}{2}}\right) & k \text{ is even} \\ \frac{1}{2(2q-k+2)} & k \text{ is odd.} \end{cases} \tag{39}$$

Combining equations (37), (38), and (39) we obtain:

$$a_k^d = C_1(d,k) \sum_{q=\lceil\frac{k}{2}\rceil}^p C_2(q,d,k) \begin{cases} \frac{1}{2(2q-k+2)}\left(1 - \frac{1}{2^{2q-k+2}}\binom{2q-k+2}{\frac{2q-k+2}{2}}\right) & k \text{ is even} \\ \frac{1}{2(2q-k+2)} & k \text{ is odd.} \end{cases} \tag{40}$$

As is proven in Thm. 3 in the paper, the coefficients for the odd frequencies in (40) (with the exception of $k = 1$) vanish.

### B.5 Coefficients with bias

Denote the harmonic coefficients of $K^b = K_3 + K_4$ by $b_k^d$ then

$$b_k^d = Vol(S^{d-1}) \int_{-1}^1 K^b(t) P_{k,d}(t)(1-t^2)^{\frac{d-2}{2}} dt \tag{41}$$

$$= \frac{1}{2\pi} C_1(d,k) \int_{-1}^1 (\pi - \arccos(t)) \frac{d^k}{dt^k}(1-t^2)^p dt \tag{42}$$

$\underline{\mathbf{k = 0}}$:

Considering $K_3$, and using (4)

$$\frac{1}{2}\int_{-1}^1 (1-t^2)^{\frac{d-2}{2}} dt = \frac{1}{2}\int_0^\pi \sin^{d-1}\theta d\theta = \frac{2^{d-1}}{d\binom{d-1}{\frac{d}{2}}} \tag{43}$$

Considering $K_4$, and using (14),

$$\frac{1}{2\pi}\int_{-1}^1 \arccos(t)(1-t^2)^{\frac{d-2}{2}} dt = \frac{1}{2}\sum_{q=0}^{\frac{d-2}{2}}(-1)^q \binom{\frac{d-2}{2}}{q}\frac{1}{2q+1} \tag{44}$$

Combining these we get:

$$b_0^d = \frac{1}{2}C_1(d,k)\left(\frac{2^{d-1}}{d\binom{d-1}{\frac{d}{2}}} - \frac{1}{2}\sum_{q=0}^{\frac{d-2}{2}}(-1)^q \binom{\frac{d-2}{2}}{q}\frac{1}{2q+1}\right) \tag{45}$$

$\underline{\mathbf{k > 0}}$:

The term associated with $K_3$ vanishes, since $(p > k-1)$

$$\frac{1}{2}C_1(d,k)\int_{-1}^1 \frac{d^k}{dt^k}(1-t^2)^p dt = \frac{d^{k-1}}{dt^{k-1}}(1-t^2)^p\Big|_{-1}^1 = 0 \tag{46}$$

Therefore,

$$b_k^d = -\frac{1}{2\pi}C_1(d,k)\sum_{q=\lceil\frac{k}{2}\rceil}^p C_2(q,d,k)\int_{-1}^1 \arccos(t) t^{2q-k} dt \tag{47}$$

where $p = k + \frac{d-2}{2}$. Replacing $t = \cos\theta$ and using (6)

$$\int_0^\pi \theta \cos^{2q-k}\theta \sin\theta d\theta = \begin{cases} \frac{\pi}{2q-k+1} & k \text{ is even} \\ \frac{\pi}{2q-k+1}\left(-1 + \frac{1}{2^{2q-k+1}}\binom{2q-k+1}{\frac{2q-k+1}{2}}\right) & k \text{ is odd} \end{cases} \tag{48}$$

Putting all this together

$$
b_k^d = \begin{cases} -C_1(d,k) \sum_{q=\lceil \frac{k}{2} \rceil}^{p} \frac{C_2(q,d,k)}{2(2q-k+1)} & k \text{ is even} \\ -C_1(d,k) \sum_{q=\lceil \frac{k}{2} \rceil}^{p} \frac{C_2(q,d,k)}{2(2q-k+1)} \left( -1 + \frac{1}{2^{2q-k+1}} \binom{2q-k+1}{\frac{2q-k+1}{2}} \right) & k \text{ is odd} \end{cases}
\tag{49}
$$

The final coefficients are given by

$$
c_k^d = \frac{1}{2}(a_k^d + b_k^d)
\tag{50}
$$

where $a_k^d$ is given in (40), resulting in

$$
c_k^d = \frac{1}{2}C_1(d,k) \sum_{q=\lceil \frac{k}{2} \rceil}^{p} C_2(q,d,k) \begin{cases} -\frac{1}{2(2q-k+1)} + \frac{1}{2(2q-k+2)} \left( 1 - \frac{1}{2^{2q-k+2}} \binom{2q-k+2}{\frac{2q-k+2}{2}} \right) & k \text{ is even} \\ \frac{1}{2(2q-k+2)} + \frac{1}{2(2q-k+1)} \left( 1 - \frac{1}{2^{2q-k+1}} \binom{2q-k+1}{\frac{2q-k+1}{2}} \right) & k \text{ is odd} \end{cases}
\tag{51}
$$