[Reviews · NeurIPS 2019]

Reviewer 1



Originality: The analysis is original to the best of my knowledge. Quality: Seems good, except a few issues that require clarifications: 1) In section 4.1 it is claimed that the eigenvalues for sine functions are zero, but in Figure 3 we see that the leading (non-zero) eigenvalue belongs to a sine funcion. How is this possible? Also, does figure 5 include sine functions? 2) In figure 5, what is the reason for the mismatch at the low frequncies? Clarity: Mostly clear except 1) What was the stopping critertion (\delta) in Figures 5 an 6? 2) In figure 5 and 6 the phrasing is a bit a obscure - it took me a while to understand what is going on with the odd frequencies (that they do not appear in the graph since they didn't reach the stopping critertion after many iterations). It is was not very clear that dots on top of left panel correspond to these non-converging odd runs. I would suggest adding a few example runs in the appendix. 3) I feel the authors should add the basic explanation why bias it is necessary for the network to approximate an odd function. It is rather simple, as I explain next. 2-layer ReLU without bias are positively homogeneous functions of x. Therefore, they cannot approximate odd functions in x. Specifically, in S^1, we must require cos(k(\theta+pi)=cos(k(\theta), so k must be odd. Significance: A nice paper with some interesting observations. Assuming my comments will be answered satisfactorily, I recommend acceptance. Minor issues: - line 149: shouldn't cos(\theta) have some normalizing scale factor? - line 198: "makes sense" not very clear. Please use more precise terms. - The equation below 198, S should be in \mathbb{}. - line 213 x^{\ell} should not be in bold. - eq. 16 "w'x" in indicator function should be "t". %%% After Author Feedback %%% The authors have answered my comments, and I'm voting for acceptance. I think these convergence results give a nice and informative characterization of the convergence rates for different frequencies.

Reviewer 2



What functions do NNs learn (approximate a function) and how fast are central questions in the study of the dynamics of (D)NNs. A common conception behind this problem is that if one trains a network longer than necessary, then the model might overfit. However, the definition of overfitting appears to vary from paper to paper. Moreover, overfitting is intimately linked with another hot topic in the area: over-parametrization. Please refer to "Advani & Saxe 2017 High Dimensional Dynamics of Gen Error for NNs" for a modern take on this link. Keeping in mind this link, we focus on fixed-size networks. If one monitors the validation performance during training, it is well established that early stopping yields better performance. Recently, Nahaman et. al. On the Spectral bias of NNs shows that NNs learn (in time) functions with increasing complexities. The present paper, "The Convergence Rate of NNs for Learned Functions of Different Properties" builds on this idea of learning functions with increasingly high frequencies. It is a well-written paper that demonstrates the core decomposition. Moreover, the decomposition yields convergence times for each component. As a major plus, the proposed rates seem to match well with the empirical observations. Even though the paper is an important step forward, it has a few shortcomings: - The mathematical description of the learning problem is limited to 1 hidden layer networks and mean square loss. It may be hard to extend proofs further. But the empirical evidence is not enough to justify its real practical value. - Authors didn't provide the code. It didn't affect the score I assigned. But it would have been a great chance to see how the curves change with different ways of scaling the meta-parameters and change the loss. - Derivations depend on (it heavily relies on [3] in the references of the paper) a particular scaling that puts the dynamics into a linear regime. There is no discussion of over-parametrization and active and lazy learning regimes. For a review and further references, please refer to Chizat & Bach 2018 On Lazy Training in Differentiable Programming. I think with the inclusions of the above discussions, code, and further experiments, it would be a very nice paper. Overall, the argument is sound and the text is clear (except for a few typos and sentence constructions). For a further point: Convergence times scale pretty bad with the dimension of the input data. But then it may depend on the intrinsic dimension of the data for real-world problems. I think it is another exciting direction that is not directly within the scope of the present work. But it would be a useful direction of research for future studies. ======= update after author response ======== Thanks very much for considering suggested changes. I hope to be able to see the code available as well!

Reviewer 3



The paper presents a theoretical analysis of the convergence of overparameterized shallow neural networks trained on regression problems. The main claim is that low frequency components of the regression landscape are learned faster than high frequency components and that this phenomenon, when combined with early stopping, leads to higher generalization performance. Major concerns: The analysis presented in the paper is definitly insightful but it is not original. The bulk of the results presented in the paper were obtained in (Sanjeev, 2019) and (Xie2017). These results include the derivation of the Gram matrix and the connection with Fourier analysis on n-dimensional hyper-spheres (Xie2016) and the asymptitic convergence results (Sanjeev, 2019) . As far as I understand, the only contribution of the present paper is in the inclusion of a bias term in the theoretical analysis which leads to a minor modification of the Gram matrix. The absence of the bias term apparently leads to the impossible to learn odd frequency components. introducing the bias term leads to a behavior for the odd frequency components that is asymptotically equivalent to the previously known behavior of the even frequency components. Consequently, the fix does not provide new theoretical insights but it simply confirms what it was previously known. Most of the conclusions drawn in the paper follows from the previous work and are therefore not original. Conclusion: The original contributions of the paper are very limited. Consequently, I believe that the paper should be rejected. References: Xie, Bo, Yingyu Liang, and Le Song. "Diverse neural network learns true target functions." arXiv preprint arXiv:1611.03131 (2016). Arora, Sanjeev, et al. "Fine-grained analysis of optimization and generalization for overparameterized two-layer neural networks." arXiv preprint arXiv:1901.08584 (2019).

[Author Response · NeurIPS 2019]

We thank the reviewers for extremely helpful suggestions. We will incorporate all of them in the final version of the
paper. Below we discuss the most significant points.

**Reviewer 1**:

**Eigenvalues for sine functions**. The kernel is an even function, and so its decomposition includes only cosine functions.
However, when it is applied as convolution, phase shifts are further included, and therefore the eigenvectors of $H^\infty$
include the sine functions with eigenvalues equal to those of the cosine functions. We will clarify this in the final
version.

**Mismatch at the low frequencies**. For the very low frequencies the actual run time is affected strongly by the
initialization, yielding slightly slower convergence times than predicted.

**Stopping criterion and fitting**. The target accuracy depends on both $\delta$ and $\epsilon$. We stopped each experiment when the
accuracy of the network reached within $5\%$ of the desired output. (We tested other values as well and obtained similar
results.) In each graph the predictions (in orange) were scaled by a single multiplicative constant to fit the measurements.
This constant reflects the length of each gradient step (e.g., due to the learning rate and size of training set).

**Clarity**. We thank the reviewer for these suggestions. We will add example runs in the appendix and include a more
intuitive explanation of the necessity of bias.

**Reviewer 2**:

**Empirical evidence is not enough to justify its real practical value**. We have added new experiments with deeper
networks (5 FC layers), different architectures (Resnet-10) and different loss functions (cross-entropy). All results are
consistent with our theory (see Figure 1).We will provide code for all experiments. We will include further results with
different hyperparameters in supplementary material; these all show similar results.

**Discussion of over-parametrization**. We thank the reviewers for this point; we agree that it merits more discussion in
our paper. We will explain that our results rely on lazy training induced by overparameterization. Chizat et al. provide
experiments that question the ability of such networks to generalize well, which seems in contrast to theoretical results
of Arora et al. and others. We will discuss this interesting question.

**Intrinsic dimension**. We agree that this is an exciting direction. As a first step, when the data lies in a lower dimensional
linear space we can show analytically that the predicted times depend on the intrinsic dimension. This is supported in
practice by experiments shown in Figure 1 (d) and (e).

**Reviewer 3**:

**Insightful but not original**. Our main contribution is to derive concrete predictions in all dimensions for the con-
vergence time of gradient descent, as a function of frequency. We do this by providing explicit expressions for the
eigenvalues of the gram matrix. This also allows us to identify instabilities in the bias-free model. Our predictions are
validated empirically on both shallow and deep networks. In this rebuttal we provide more general experiments for
different architectures and loss functions. Rev. 2 states: "As a major plus, the proposed rates seem to match well with
the empirical observations". All of these contributions go beyond [Xie 2017 and Arora 2019], which are cited and
discussed in the paper.

We also feel that our discussion of the role of bias goes beyond a "minor technical fix". It is clear from prior work that
$H^\infty$ might contain eigenvectors with small eigenvalues, corresponding to functions that are hard to learn. However,
that work did not show that these functions are intuitively complex; this was not possible because this is not true for
networks without bias. By introducing bias, we are able to show for the first time that hard-to-learn functions in fact do
correspond to high frequency functions.

Figure 1: Number of iterations to convergence as a function of target frequency. From left to right: (a) input in $S^1$, MSE loss, 5 layers fully connected; (b) same, 10 layer Resnet; (c) Resnet-10, cross entropy loss; (d) input in $S^1$ embedded with a random rotation in $\mathbb{R}^{30}$, MSE loss, Resnet-10. (e) input in $S^2$ otherwise same as (d). For (c) the task is binary classification; so that for every $k$, class is 1 if $\cos(k\theta) > 2/3$, -1 if $\cos(k\theta) < -2/3$, training points with other $\theta$ values are not used. Consistent with our theory's predictions for two layer networks, we see quadratic growth in cases (a)-(d) and cubic growth for (e). To estimate the leading exponent in these graphs we fit a line to the corresponding log-log plots, obtaining, from left to right, $O(k^{1.93}), O(k^{1.95}), O(k^{2.37}), O(k^{2.08}), O(k^{3.09})$.

[Meta-Review · NeurIPS 2019]

This paper analyzes the convergence rate of single-hidden-layer ReLU neural networks along different Fourier frequencies. It finds that lower frequencies learn first, and finds that biases allow for learning of odd frequencies. The restriction to spherical data is limiting, but the analysis and conclusions (particularly the rates of convergence) are novel and interesting. I recommend acceptance.